# The Role of Mass Media in Influencing the Lifestyle of the Elderly during the COVID-19 Pandemic

**DOI:** 10.3390/healthcare11131816

**Published:** 2023-06-21

**Authors:** Lilia Sargu, Felicia Andrioni, Lavinia Popp, Adrian Netedu, Maria Cristina Bularca, Adrian Otovescu, Gabriela Motoi, Ion Negrilă, Cosmin Goian, Claudiu Coman, Gianina Chirugu

**Affiliations:** 1Department of Economy and Tourism, Faculty of Economic Sciences, University of European Studies of Moldova, 2069 Chisinau, Moldova; lsargu@mail.ru; 2Department of Socio-Humanities Sciences, Faculty of Sciences, University of Petrosani, 332006 Petrosani, Romania; feliciaandrioni@upet.ro; 3Department of Social Work, Faculty of Sociology and Social Work, University “Babeş-Bolyai”—Reşiţa University Center, 400347 Resita, Romania; lavinia.popp@ubbcluj.ro; 4Department of Sociology and Social Work, Faculty of Philosophy and Socio-Political Sciences, Alexandru Ioan Cuza University of Iasi, 700506 Iasi, Romania; adrian.netedu@uaic.ro; 5Department of Social Sciences and Communication, Faculty of Sociology and Communication, Transilvania University of Brasov, 500036 Brasov, Romania; cristina.bularca@unitbv.ro; 6Department of Communication, Journalism and Education Sciences, Faculty of Letters, University of Craiova, 200585 Craiova, Romania; adrian.otovescu@edu.ucv.ro; 7Department of Sociology, Philosophy, and Social Work, Faculty of Social Sciences, University of Craiova, 200585 Craiova, Romania; gabriela.motoi@edu.ucv.ro; 8Doctoral School of Social and Humanities Sciences, University of Craiova, 200585 Craiova, Romania; negrila.ioan.t2u@student.ucv.ro; 9Department of Social Work, Faculty of Sociology and Psychology, West University of Timisoara, 300223 Timisoara, Romania; cosmin.goian@e-uvt.ro; 10Faculty of Theology, Ovidius University of Constanta, 900527 Constanta, Romania; geanina.chirugu@365.univ-ovidius.ro

**Keywords:** elderly, quality of life, pandemic, resilience, coping mechanisms

## Abstract

The elderly represent one of the categories that was most affected by the pandemic period. The purpose of this research was to analyze the ways in which mass media and very often contradictory information flows influenced the lives and personal communications of the Romanian elderly population during the COVID-19 pandemic. In order to conduct the research, we used a mixed-methods approach. For the quantitative research, we gave a questionnaire to the elderly population of Romania, and for the qualitative research, we conducted interviews. Our quantitative sample included 881 retired persons with an age between 55 and 94 years old; the mean age was 71.48 years old with 6.6 years standard deviation. The elderly described the pandemic period using negative words: fear, loneliness, anxiety, disaster; the channel they mostly watched and trusted was the TV; they were aware of the measures they had to take to protect themselves; they missed most of the meetings with the family and the main problems they had were represented by loneliness, the inability to be with their families or the lack of access to medical services. The elderly’s mass media consumption during the pandemic was mostly represented by TV consumption; the information spread by mass media was sometimes contradictory; it influenced their behavior and may have generated feelings of anxiety among them.

## 1. Introduction

At the level of the European Union, the demographic aging process has registered an upward curve in the last two decades due to some contextual factors of increasing the quality of life of the population and, for this reason, the share of elderly people in the total population is quite high. The situation is similar for Romania, the statistical situations provided by [1] highlighting the fact that 19% of the total population is represented by the over-65 population category [2,3].

The aging process involves two essential aspects that are closely related: on the one hand, biological aging and, on the other hand, quality of life [4]. Some significant elements illustrated by [5] for the quality of life of the elderly are functional capacity, intergenerational relationships, health status, standard of living, social support, independence status, the environment in which they live, spirituality and state of mental health [6]. The stated elements present a different degree of importance for the elderly and, depending on their individualized needs and problems, the elements of the quality of life of this category can be perceived differently [2]. A study highlights the fact that the health status of the elderly population in Romania is weaker compared to other categories of the elderly population in other more developed European states [2].

Humanity has been subjected to unprecedented challenges, including the recent SARS-CoV-2 pandemic, even though, over time, appropriate treatments or solutions have been identified for many infectious diseases [7]. The SARS-CoV-2 pandemic has generated several risks and difficulties globally, with the onset of this pandemic reported in China in 2019 and the rapid spread of the pandemic to 118,000 cases globally in 114 countries after only 2 months of its emergence [2,8,9]. The first confirmed case in Romania was registered on 26 February 2020 [10,11] and by 1 November 2021, 1,648,031 Romanians had been diagnosed with coronavirus, from which 47,751 died and 1,405,694 recovered [12,13].

The most exposed age group to illnesses was the age group between 50 and 59 years old, highlighting the fact that over 85% of the deaths in Romania during the pandemic period were registered in elderly people over 60 years old with comorbidities [14].

Romania was extremely affected by the COVID-19 pandemic situation in terms of the rate of illness and mortality per 100 thousand inhabitants [15]. Thus, the COVID-19 pandemic constituted one of the major challenges at the global, European and national levels. A social category extremely affected by the pandemic generated by COVID-19 was the category of elderly people. The present work brings to the fore the elderly, their characteristics and lifestyle during the pandemic and how the mass media had a major influencing role during the pandemic that influenced the lives of these people. Hence, the purpose of our paper was to analyze the ways in which mass media, and very often contradictory information flows, influenced the life and personal communications of the Romanian elderly population during the COVID-19 pandemic.

The complexity of this work is determined, on the one hand, by the theoretical approach to some aspects related to the pandemic situation generated by COVID-19 in Romania and its implications for the elderly in various aspects of life, to the aging process, to the existing multiple stressors and implicitly understanding the resilience capacity and coping mechanisms of the elderly during the pandemic period, to the role of the mass media in influencing the quality of life of the elderly. On the other hand, the applied research carried out at the national level completes the information, bringing added value to this field with a quantitative–qualitative mixed analysis and interpretation part that illustrates how the media communication channels have left their mark on the various aspects of the life of the elderly, the perception of the elderly on their own mental and emotional health or social relationships and how they were affected by the pandemic generated by COVID-19. In addition, another important aspect monitored was the perception regarding the life of the elderly in the pandemic period, in family, social and medical terms and the problems encountered, as well as the methods of support they received. On 2 November 2021, Romania was the second-least vaccinated state of the European Union, even though a record number of daily deaths from the coronavirus were reported and there were no intensive care beds available in Romania [16].

The emergence of COVID-19 has resulted in an unprecedented disruption of both physical and mental health among the world’s citizens. In [17,18], among other effects generated by this, the upward dynamics of social uncertainty and distrust in public institutions are highlighted [2]. A broader time perspective is needed to perceive the effects of the pandemic on the population [19]. The pandemic situation caused the emergence of a social and medical crisis with major implications for the vulnerable category of the elderly [7].

Regarding the measures to prevent the spread of the virus, Romania had to apply a strategic plan to control and prevent the expansion of the pandemic situation either through certain measures to restrict access to air flights, the closure of certain public or private institutions and large stores, the mandatory self-quarantine of affected persons being introduced or other preventive hygiene measures to prevent the spread of the virus [11].

In order to prevent the spread of the virus, and at the level of elderly beneficiaries of social services, various legislative measures were applied to restrict access to these services [7]. Another challenge for Romania was generated by the numerous demands of the elderly for care in social services caused by social, medical and economic factors: low incomes and pensions, high costs for medicines and food and increased costs of maintenance expenses [20]. Other categories of social services affected by the pandemic situation were day, residential and home social services intended for the elderly, who had to readjust their intervention plans, procedures, methodologies or activities concerning the new crisis conditions generated by the pandemic [21,22]. The consequences of COVID-19 in residential social services have been devastating. Countries such as the United States, Spain and Italy had a high mortality rate among institutionalized elderly. The proximity between residents led to a higher risk of infection and, consequently, adverse outcomes and mortality associated with difficulties in preventing the spread of the virus in institutions [23,24,25]. Clinicians and caregivers were prompted to estimate the availability of resources and consider how the absence of resources can be mitigated for a given individual and family. Of particular importance was the role of technology, which emerged as an important factor in maintaining social connectedness as well as accessing mental health services [24].

Several studies [26,27] demonstrated that the older generation [28] was affected to the highest degree compared to other vulnerable groups due to the higher risk of isolation, self-isolation and social exclusion associated with the high risk of contracting the disease and losing life [22,29,30,31,32,33] or with the risk of discrimination and manifestation of “ageist” attitudes [34,35,36]. The factor of aging correlated with the factor of COVID-19 is worrisome, considering where studies present the risk of minimizing the value of an older adult’s life, both among healthcare providers and among the public [37]. These findings are troubling, but also form the basis for understanding elder abuse. Rates of elder abuse have increased since the onset of the pandemic, and this type of abuse has been and is often unreported [38].

The impact of the severe acute respiratory syndrome (SARS) on mental health [39] among the public has been previously reported [40], and recent systematic reviews are beginning to highlight the detrimental impact of COVID-19 on mental health among different populations [41,42,43]. Social isolation measures are a “perfect storm” for mental illness [44,45,46], in the elderly [47]. Their daily social activities, such as walking and socializing, meeting friends, volunteering and religious gatherings, were restricted due to quarantine orders and transport controls. This exacerbated the challenges of the pandemic and had a severe impact on their mental health [48,49].

The COVID-19 pandemic induced worry [50], fear [51], anxiety and depressive symptoms [52] as well as insomnia [53]. While social distancing has shown positive effects in terms of the number of infections [54], the sociopsychological consequences are quite discouraging. Social isolation increases the risk of mental health problems [55,56]. Policy measures calling for social distancing could therefore directly affect feelings of loneliness and depression [45,57]. In particular, the elderly and people living alone, who appear to be even more vulnerable to mental health problems, could be affected by these developments [58,59,60]. Mental health problems themselves are known to be associated with an increased risk of long-term morbidity and mortality, particularly for older people [61]. Moreover, social isolation in the elderly has been shown to lead to increased risks of cardiovascular [62], autoimmune and neurocognitive [63] disease. Fear of contracting the disease [64] and losing loved ones constituted mental stress and generated anxiety in the elderly population [65]. The negative impact of COVID-19 on mental health in the general population has been identified as a research priority [42,45,46,66,67]. A study by [68] reported a high prevalence of generalized anxiety disorder and poor sleep quality in Chinese individuals during the COVID-19 outbreak. Another study [69] observed a higher prevalence of anxiety and depression in people affected by quarantine compared to those not affected by China’s quarantine measures in early February 2020. Krendl and Perry focus on older adults in the US [70] and find that their mental health has been negatively affected by the COVID-19 pandemic. Respondents experienced higher levels of depression and loneliness than they had before the pandemic. Van Tilburg et al. found that the level of loneliness among elderly people in the Netherlands increased, while overall mental health remained roughly stable [71]. The study shows that social distancing measures were not the main determinant for the increase in the prevalence of loneliness [72].

In general, the socio-economic conditions that were imposed on the people en masse, the forced lockdown without provision of basic human needs, poor governance, communication, infrastructure and healthcare facilities, created public anxiety [55] and disturbed human life [73]. Given the immense burden of COVID-19 on older populations, it is critical to understand what have been effective coping strategies [74]. Older adults use various coping strategies to deal with this crisis that draw our attention to the dynamic, changing and temporal dimensions of the coping process, especially regarding expectations about the duration of stressors as a way to enable the population to adapt to stressors [75].

The deeper the interpersonal connection and interpersonal involvement, the more it favorably influences human resilience, and at the opposite pole, the state of fear and lack of certainty as well as loneliness represent impediments to reaching a reasonable level of resilience or in reaching individual well-being and health [76].

Social engagement, which includes meaningful interactions and connection with others, is an essential stimulus needed for older people to improve their physical and mental health as individuals [77,78]. Positive social connections and relationships are considered to be an integral factor in the well-being of older people as social beings [79].

Therefore, in the presence of this social imbalance, older adults are challenged to develop strategies to cope with such a situation [77,80]. These coping strategies are natural coping capacities known as internal resilience [81] developed by an older person in response to the aging process and stressful life experiences such as the COVID-19 crisis [77].

Coping strategies associated with managing the stress and anxiety of social isolation related to COVID-19 have not yet been widely studied [82], especially among older adults [83].

In reconfiguring a path of personal balance in the risk conditions generated by the stressors during the COVID pandemic period, it was necessary to draw up a coping strategy [84]. Depending on individual adaptive resources, negative stressors can be counteracted and, implicitly, the activation of adaptive mechanisms to adverse events or illness [85] or to the lack of self-efficacy [86].

Each individual has a unique coping strategy in relation to their ability to psychologically adapt to the disease situation that leads to the management of the stressful situation [87,88,89,90,91].

The stressor or risk factor in the present situation is the fear of infection with the COVID-19 virus. Coping ability is the central process in building resilience. The identified stressor is dangerous, so coping responses are activated. Resilience is the result of the interaction of elderly people with their environment and the processes that promote well-being or protect them against the overwhelming influence of the pandemic [92]. These processes can be individual adaptation strategies or coping mechanisms or they can be stimulated and facilitate the emergence of resilience.

The notion of coping, currently used extremely frequently by specialists, signifies the association between an individual and the contextual life situation, an association that can lead to stress reduction but also signifies the mechanism that generates adaptation to the stressor [93]. This coping mechanism aims at the transformation and adaptation of systemic segments to be able to respond adaptively to dynamic stimuli generated by external environmental factors [94].

Lazarus and Folkman define the notion of coping as the behavioral or cognitive ability of the individual to manage external or internal demands in relation to the availability of individual resources [85]. It is an adaptive reaction in response to actual or imagined stress to maintain psychological integrity [95]. Within this concept, people are perceived as responding to stress through attack, fight or compromise reactions. These reactions become complicated due to the numerous coping mechanisms whenever the stressor involves the Ego.

Normally, defense mechanisms are used constructively in the coping process. Available coping mechanisms are what people usually use when they have a problem. The Great Pandemic of 2020 was a unique stressor that affected communities around the world [39]. The nature of this contagion virus has led to mass fear and hysteria and many misconceptions, with widespread stigmatization and discrimination against those infected. Altogether, these perceptions have created immense stress among the population, but still, they are trying to cope with this situation. They try to learn coping strategies to adapt to these traumatic crises. To cope with a situation, a person tries to manage their external and internal demands that are perceived as demanding or exceeding adaptive resources [96].

Coping acts as a stabilizing factor that can help the individual maintain psychological adaptation during a stressful period [97]. Coping mechanisms can be divided into two categories: adaptive coping and maladaptive coping. The adaptive coping strategy usually benefits or positively affects the lives of those who use it. Ref. [98] and some examples of this approach include religious/spiritual coping, such as prayer and scripture reading [99]; exercise [100]; meditation; music auditions and socializing with friends and family [101].

The other form of coping, maladaptive coping, refers to methods that often lead to adverse consequences, including some mental health challenges. Previous research has divided maladaptive coping into two different categories: emotional, where individuals respond to a situation confrontationally or with an excessive emotional response, and avoidance-based, where individuals actively delay responding to a stressful situation or escape a situation altogether through isolation or other maladaptive behaviors [98,102,103]. Maladaptive behaviors may include alcohol consumption [104], tobacco [105], drug use and overeating [101] and other risk behaviors [106]. Historically, these coping strategies can lead to negative effects on one’s life, including greater depression, greater pain and functional impairment, and lower self-efficacy [107]. Thus, coping mechanisms can focus on the meaning, practical aspects, emotional aspects and avoidance aspects of a crisis. People differ greatly in their ability to successfully cope with stressful life events, and some coping strategies are much more effective than others when faced with a selected life-changing event [108]. How one deals with a crisis has important implications for well-being. Effective coping leads a person to achieve better life satisfaction.

As for coping, it involves a process and several anticipatory, confrontational and analytical stages that lead to the reduction of the stressful situation. Therefore, coping can represent a stable personal predisposition [109], but it is also seen as an evolutionary process, including genetically [110,111]. Theoretical perspectives on coping suggest that older age may be advantageous for coping due to the accumulation of life experiences that encourage the refinement of coping skills [112,113]. The process of coping with stress can build resilience, and for older adults, there is an increased likelihood that they have increased coping resources and strategies to draw upon from their experiences [112,114].

The COVID-19 pandemic presents a variety of stressors, including social distancing, but it is conceivable that older adults will apply their accumulated coping skills to cope. Coping strategies are essential factors that determine one’s well-being in old age [115]. Coping strategies compensate for or mitigate stressful circumstances by reformulating and adapting to a new and positively evaluated environment [81,115].

Internal resilience theory proposes that in times of crisis, health needs to compel older people to develop internal resilience to preserve their integrity, well-being and quality of life as individuals [81]. Internal resilience refers to the ability to adapt well in the face of adversity, danger, trauma or significant sources of stress, collectively known as a crisis, to sustain a sense of purpose and vigor and to emerge stronger from such stressful situations [77,81].

Regarding the forms of coping, a classic classification is delimited into problem-centered coping and emotion-centered coping [85].

Emotion-focused coping is generally used by older people to adapt to stress [116].

Coping is illustrated by Stone and Neale as being characterized by several factors such as acceptance, social support, direct action, relaxation and religious feelings [117], and Folkman and Lazarus also complete the palette of specific factors with aspects related to self-control, distancing, confrontation, planning to remedy the problem or positive re-evaluation [118].

Another study [84] proposed a coping framework that describes the interconnections between characteristics of how individuals cope. Personal resources include characteristics of the individual such as dispositional coping style and characteristics such as cognitive and intellectual ability and general personality traits. These systems influence a cascade of processes that affect the psychosocial functioning and well-being of the individual. Thus, the authors of [84] describe coping in two domains: coping styles and coping responses. Coping styles are relatively stable and describe how individuals typically interact with their environment, while coping responses are used by the individual to manage particularly stressful encounters [80,84].

Another study recorded the notion of locus of control, which he associated with those traits by which individuals express their success or failure through controllable or uncontrollable causes [119]. The belief that the responsibility for success or failure is determined by the qualities or defects of the individual was named LCI-internal locus of control; in the same context, LCE was also identified, which signifies the belief that fate has a significant role in generating favorable or unfavorable events of life [120]. The meaning of self-efficacy (AE) is also illustrated, as well as the belief in the mobilization of motivational and cognitive resources in the successful performance of certain tasks [110]. The sense of coherence is structured from childhood and adolescence and is consolidated around the age of 30 and has three components: comprehension, control and purpose, and self-esteem. Thus, self-esteem correlates positively with adaptive coping strategies such as refocusing towards planning, refocusing positively and positive reappraisal and correlates negatively with maladaptive coping strategies such as self-blame, catastrophizing, rumination and blaming others [121]. Optimism is a personal resource defined as a general expectation or belief that good things will happen frequently in the future, while bad things will happen only rarely [122]. Perceived social support is defined as the perceived availability of support if needed. Studies indicate that optimism and perceived social support are effective resources for coping with adversity and challenges and may serve as a primary source of self-care and subjective well-being, especially in old age [123,124]. These latter variables are known to decrease well-being and increase the risk of depression and cognitive dysfunction. Another study pointed out that a decrease in activity and mobility in older adults during isolation can also lead to more frailty and lower well-being in older adults. Moreover, in response to stress, sleep quality may decrease and increase the risk of depression [106].

Depression may be negatively associated with health-related quality of life. Studies have found that COVID-19 patients with depression reported lower health-related quality of life compared to those without depression [125]. Similar findings have been found for other diseases such as diabetes [126] and gastrointestinal diseases [127]. The COVID-19 pandemic period has accelerated the risk and manifestations of depression, with depression influencing the ways to remedy or counteract individual problems or generating difficulties in achieving personal or professional goals, making recovery from the crisis even more difficult. Even if it manifests itself in different ways, at the base of depression there is always an attitude of giving up [128].

Based on the above analysis, understanding the implications of COVID-19 restrictions on the health and well-being of older adults is of major importance [129].

Depression is a common condition in older adults, being the second cause of global morbidity after cardiovascular diseases, and also having multifactorial origin. Depression has diverse clinical manifestations and is difficult to identify, which causes a decrease in the quality of life and an increase in suffering, often leading to fatal outcomes [130,131,132].

Coping involves cognitive and behavioral strategies that individuals use to cope with or control stressful circumstances and can be influenced by several biological and psychosocial factors, including physical health, personality, spirituality and social support. Active coping involves behaviors to proactively address, modify or overcome a stressor or situation. Regulatory coping refers to reflecting on the stressor to reduce its effects, such as reframing a stressor or adapting through a change in attitude, expectation or perception. An individual may consciously or subconsciously use both types of strategies simultaneously [71,133]. Not all coping strategies are successful or helpful. Denial, for example, may not be the most appropriate response to a problem, but it is a cognitive coping strategy frequently used as people age [134]. Aldwin’s five main categories [135] are problem-focused coping (behaviors and cognitions oriented toward solving or managing a problem, such as implementing a plan), emotion-focused coping (managing the emotional reaction to the problem), social support coping (eliciting helping others or offering support to others), religious coping (seeking help from a higher power, such as through prayer) and cognitive reframing (trying to understand the problem and/or focusing on the positive aspects of the situation). Resilience is a multifaceted and important concept: high resilience in later life has been associated with reduced risk of depression and mortality, better self-perception of successful aging, increased quality of life and improved lifestyle [136,137,138].

The mass media has an important role because it must provide correct information about the coping resources available to the elderly. The mass media can prepare the public to face the disaster, it can give warnings and information about coping strategies, it can provide a forum for public reactions and it can provide a report of the events. The substantial stress of “information overload” can lead to paranoia and mistrust of health care, which could lead one to avoid quarantine, with serious consequences for public health [139].

Thus, the 2019 coronavirus (COVID-19) pandemic was a public health crisis that required interventions to reduce the spread of the disease, including social distancing and isolation [140], generating stressful situations that challenged individuals’ coping skills. As older adults have been particularly encouraged to practice social distancing and self-quarantine, given their higher risk of hospitalization and mortality from COVID-19 [141], it is important to understand how they adapted and coped with this social isolation [83]. It is possible that, given their wealth of life experiences, older adults are uniquely able to adapt well during this unprecedented crisis [142].

Moreover, recently published studies have also suggested that COVID-19 had a particularly significant impact on psychological well-being, including suffering from social deprivation due to isolation or quarantine, feeling stressed, experiencing anxiety [143] or depression, or suicide in more serious cases [144].

However, the consequences of self-isolation, such as separation from loved ones and friends, loss of usual routine, participation in fewer social activities and limited access to regular medical care can be distressing for individuals [78].

Overcoming this situation requires the identification of effective factors in controlling and combating this epidemic in order to increase resistance to the virus and contribute to the social, economic and social development of the community [145].

### Variables Taken into Consideration in the Research

Considering the purpose of our paper, in our research we examined variables that referred to the sources of information of the elderly and their way of living during the pandemic. In this regard, the variables we examined refer to: type of media channels followed, type of information they have seen about the virus, people with whom the elderly talked most often about the virus, their trust in media channels, ways through which they verified information regarding the virus, ways of fighting/coping with the pandemic period, perception about their mental and emotional health, the means of communication that the elderly used in order to talk with friends and family members, the topics they discussed most, the main problems they have faced during the pandemic and the medical and social support they have received in the pandemic period.

## 2. Purpose and Objectives of the Research

The purpose of this research was to analyze the ways in which mass media and very often contradictory information flows influenced the life and personal communications of the Romanian elderly population during the COVID-19 pandemic.

Related to this purpose stated above, we also formulated a series of objectives.

Main objectives:-Identifying the ways in which the elderly used media channels and personal networks during the pandemic to find out about the evolution of the pandemic and the main types of information accessed.-Identifying the level of trust in the information broadcast by media channels.-Identifying the decisions made by the elderly regarding the preservation of health as a result of exposure to media channels, the preservation of personal support networks and the avoidance of social isolation.

Secondary objectives:

O1. Identifying the media channels used by the Romanian elderly to get information about the pandemic situation.

O2. Identifying the main types of information followed by the elderly on the media communications channels.

O3. Evaluating the level of trust in the information transmitted by the followed media channels.

O4. Identifying the decisions the elderly made in order to maintain their health after they had been exposed to information from the media about the virus.

O5. Evaluating the perceptions of the elderly regarding how the pandemic affected their health or mental or emotional state.

O6. Identifying the themes and ways in which the elderly communicated with relatives or acquaintances regarding the effects of the pandemic.

O7. Evaluating the extent to which the social relations of the elderly have been affected.

## 3. Hypotheses of the Research

Related to the quantitative research, we formulated the following research hypotheses:

**Hypothesis 1** **(H1).***The use of the media channels to receive information about the pandemic was significantly different with gender, residency, age category and study*.

**Hypothesis 2** **(H2).***There will be a statistical difference in trust for information transmitted about the virus and the pandemic situation between TV channels, radio, newspapers, social networks and news channel sites*.

**Hypothesis 3** **(H3).***The level of trust in media channels was significantly different with gender, residency, age category and study*.

**Hypothesis 4** **(H4).***The evaluation of “satisfaction with socialization” is greater than the “fear of isolation” among the elderly from our sample*.

## 4. Materials and Methods

### 4.1. Data Collection Method

We used for this research a mixed methods design, named more precisely *convergent parallel design* [146]. These designs refer to the implementation of quantitative and qualitative techniques at the same time, both being equally prioritized.

In our quantitative research, we gave a questionnaire to a representative sampling of the elderly population. The main selection rule was to select noninstitutionalized pensioners to constitute a representative sample at the level of Romania after a weighting according to the eight development regions of Romania. In every region, the pensioners were randomly selected after a new weighting on the urban and rural environments. A small part of these pensioners is younger than 65 years (but for various reasons they can retire earlier). Thus, we conducted exploratory research. The questionnaire contained open and closed questions formulated as concisely and directly as possible. Although in the specialized literature, there are numerous validated scales used in research on the elderly, we preferred a questionnaire that combines several questioning techniques. The questionnaires were completed face-to-face by the operators and the application period was on average 20 min. The randomly chosen respondents were interviewed in public places and not at their homes. Hence, the participants were recruited individually, not in groups, meaning that the participants were addressed individually, being asked to fulfil a questionnaire. In terms of the sample size we established, for our cross-sectional study we established the sample size according to well-known calculation methods [147]. These authors and others calculated the size of each sample and the maximum allowed errors (provided that the size of the researched population is not included in the calculation formulas). We followed a sample volume that could be balanced in terms of costs and benefits. From the beginning, we considered a sample of a maximum of 1000 respondents, but after application in the field, the elimination of incomplete questionnaires, etc., the final volume was 881 questionnaires. For this sample, the margin of error was +/− 3.3% at a confidence level of 95%.

In terms of the eligibility of the participants, [148] eligibility refers to “whether or not a sampled unit is eligible to have data gathered from it”. For our research, we considered that the entire population was affected to a greater or lesser extent by the COVID-19 pandemic regardless of age category. In this context, any person chosen at random in the sample with the age of at least 55 years is considered legitimate to enter the sample, on a voluntary basis and in compliance with the ethical norms of sociological research in general.

The operators distributed sampling quotas according to gender, age categories and living environment. Participation in the study was voluntary. The respondents were guaranteed the protection of their personal data, and anonymity and confidentiality were ensured (identification data, email addresses, telephone numbers, etc., were not collected). Respondents could also withdraw at any time after completing the questionnaire. Considering the topic of the research, the number of refusals was very low: just a few random incidents. Before agreeing to answer, a number of respondents wanted to know if the questionnaire was on political topics, a fact that would have led to a large number of refusals. However, the operators were forced to refuse people who at a given moment did not fit into the distributed quotas. Thus, the exclusion criteria referred to the fact that the participant did not want to answer the questionnaire or they did not meet the quotas established. The respondents were chosen from all the eight development regions of Romania. Data collection was carried out through student operators from the partner university centers. Considering the qualitative research, we used the snowballing method to gather potential respondents for our research.

### 4.2. Sample

The research was conducted between February and June 2022. Our sample was representative at the national level and included retired persons with an age between 55 and 94 years old; the mean age was 71.48 years old with 6.6 years standard deviation. The subjects were selected by stratified random sampling from all eight Romanian development regions (usually, to the seven large development regions of Romania, the region formed by the capital Bucharest and Ilfov county is added).

The final volume of the sample was 881 people, the maximum error was ± 3.3% and the confidence level was 95%. The structure of the sample is described in Table 1. Considering the qualitative research, we selected 193 elderly people with whom to conduct interviews from the eight development regions of Romania and the respondents were randomly selected.

### 4.3. The Research Instruments

In terms of the quantitative analysis, the research instrument used was the questionnaire. The questionnaire developed contains three sections: section A contains questions that refer to the way the elderly have obtained information about COVID-19 during the pandemic, section B contains questions that refer to the lifestyle of the elderly during the pandemic and section C contains questions regarding sociodemographic data. The questionnaire can be seen in Appendix A.

In terms of the qualitative research, we used as an instrument a semistructured interview guide. The interview guide comprised the following dimensions: general questions about the pandemic, the channels most used by the elderly to obtain information, the persons with whom the elderly talked the most during the pandemic, problems that the elderly encountered, medical services to which they had access to, social services to which they had access to, family life, relationships with their community and sociodemographic data. The interview guide can be seen in Appendix B.

Hence, the analyzed variables were: a word describing the mental representation of the term COVID-19, most followed media channels, most followed television channels, types of information followed, the frequency of discussions about the spread of the virus, the frequency of discussions about disease prevention, categories of interlocutors about the pandemic, the level of confidence in diverse media channels, forms of validation of the information, decision made after the media exposure, effect of the pandemic on health state, effect of the pandemic on mental state, emotional states during pandemic, effect of distancing measures, categories of relatives in direct contact, intensity of communication with relatives, types of channels of interpersonal communication, pandemic related subjects of communication, intensity of social relationship, intensity of necessary support received, the quality of family ties, the tendency to avoid social contacts, the intensity of the feeling of alienation, main activities during the pandemic, main life aspect affected, type of living, possession of a dog, the level of satisfaction with the vaccine, reasons for vaccination, reasons of nonvaccination and main challenges during the pandemic.

### 4.4. Data Analysis

The data collected were analyzed with the 20th version of the program Statistical Package for the Social Sciences (SPSS).

In our analysis, we included as predictors gender, residency environment, living place during the pandemic, age and level of education. To test the hypothesis we used: the Chi-Square Test of Independence (for Hypothesis H1), the Freeman nonparametric test for k related sample, the Wilcoxon nonparametric test (for Hypothesis H2), the construction of statistical index “trust in media”, nonparametric Mann–Whitney U test and the Kruskal–Wallis Test (for Hypothesis H3), construction of two statistical indexes “socialization” and “isolation”, a nonparametric Wilcoxon test and Spearman correlation (for Hypothesis H4).

## 5. Results

### 5.1. Results of the Quantitative Research

#### 5.1.1. The Elderly and Sources of Information during the Pandemic

From the beginning, we invited the respondents to specify the first word that comes into their minds when they think about the COVID-19 pandemic. After recodification, we obtained the results presented in Figure 1.

The terms specified by the respondents are surprising in terms of their harshness and negativism. Thus, the main words that the elderly used to describe the pandemic were: fear, disease, restriction/quarantine/medication, etc., and these terms alone suggest the major difficulties that affected this category of people. Among these terms, the problem of “fake news” appears, although with low percentages. Moreover, a simple search on the Internet can reveal a multitude of fake news with direct reference to elderly people.

Next, according to objective O1, we tracked which were the main media channels consulted to obtain information about the pandemic. The vast majority first specified “television” (81%), which was followed by low scores “the online environment/Internet” (9%), “radio” (5%) and “journals” (3%). We would like to note that television was indicated as the main information media channel in most of the sociological research conducted during the pandemic in Romania.

Regarding the current use of the Internet by the elderly, we have information from a previous study [149], only 30% of people aged 65 overuse the fixed Internet and only 34% have the Internet available on their mobile phone Among the Romanian elderly population who connect to the Internet, the main activities are presented in Table 2.

Starting from the data in Table 2, we state the following hypothesis:

**Hypothesis 1** **(H1).***The use of the media channels to receive information about the pandemic was significantly dif-ferent with gender, residency, age category and study*.

The statistical analysis showed that there are no statistically significant differences regarding the use of different media channels in terms of gender (Chi Square = 7.76, df = 4, *p* ≥ 0.05) and residency (Chi Square = 4.58, df = 4, *p* > 0.05). We concluded that the respondents similarly use the media channels regardless of the gender of the subjects and the residential environment (urban or rural). Instead, we determined that there are statistically significant differences regarding the use of different media channels in terms of “school completed” (Chi Square = 30.72, df = 12, *p* = 0.002) and age category (Chi Square = 32.67, df = 12, *p* < 0.001). In both cases, the effect size was low (phi = 0.186, *p* < 0.01 and phi = 0.192, *p* < 0.01) and we conclude that the association between variables is significant but with low intensity. From the cross-tabulation tables, we concluded that respondents with a high school education level have significantly higher media consumption than the other respondents. In the second case, we observed that the respondents aged between 65 and 70 have significantly higher media consumption. Hypothesis 1 (H1) is partially confirmed. Considering that television turned out to be the main media channel, we were interested in what were the main TV stations that were watched during the pandemic to get trustworthy information. According to the results of the research, the channel most watched by the respondents was Antena 1 (25.6%), followed by Pro TV (16.2%) and Realitatea TV (15.3%). These three channels are the news channels most popular in Romania. In this case, it was possible that confident media channels could distort, voluntarily or not, information about the pandemic. We asked the respondents to characterize the perceived intensity of some types of information through media channels. The percentages are visible in Table 3.

We noted that preventive information was dominant (a, b) followed by evaluative information (c, d) and the last three variants (e, f and g) were the medical information, unfortunately at the end of the list. In the previous table, 68% of the respondents declared that they have often and very often encountered positive information about the vaccine, while 48% have encountered negative information. Both types of information were circulated abundantly during the pandemic and it is difficult to decide to what extent such information was decisive in the decision to be vaccinated or not vaccinated. In Table 4, for example, we presented the case of those who decided to get vaccinated and the fact of being informed about the vaccine.

The data in Table 4 were obtained by cross-tabulation between variables A4 and A10. We observed that the respondents who made the decision to get vaccinated were exposed to both types of reports about vaccination, positive or negative. Next, we wanted to find out if there was any correlation between the behavior of the elderly of talking to someone about the spread of the virus and of talking to someone about methods of prevention (Figure 2).

From this graph, we understand some similarities between the answers to these two questions. This relation is proved by a Spearman correlation between these two variables (Rho = 0.639, *p* < 0.001). In conclusion, the discussions about the spread of the disease correlate with the discussion about disease prevention. These remarks can be developed in other qualitative research in terms of the role of maintaining interpersonal communication even in the condition of isolation in exchanging personal experiences and advice at the beginning of a pandemic period dominated by many unknown factors: preventive behavior, validation of information, sharing experiences, etc. Numerous research studies have insisted on the role of communication during the pandemic in diverse directions: controlling misinformation [150], masking effects [151] and health-protective behaviors [152]. Next, we were interested in finding out with whom have the elderly mostly exchanged information about the virus. The results are presented in Table 5.

The results of the research show that the elderly communicated the most with their children (42%) and their spouses (23%). However, some of them preferred to talk to other family members (13%), to their friends (10%), to their siblings (5%) or to neighbors (4%) (Table 5). Regarding the confidence in information spread about the virus by the media channels, it was important to establish the significant statistical differences in trust between the stipulated media channels (TV channels, radio, newspapers, social networks, and news channel sites). The first descriptive impression we can have calculated is the mean of the Lickert scale in all cases For example, the means and std. deviations for media channels were very close: for TV channels (m = 3.99; SD = 1.62), for radio (m = 3.65; SD = 1.58), for newspapers (m = 3.46; SD = 1.61), for news channel sites (m = 3.04, SD = 1.57) and for social networks (m = 2.99, SD = 1.59). Starting from these data and the eventual statistical analysis about significant differences between means we enounced the next hypothesis:

**Hypothesis 2** **(H2).***There will be a statistical difference in trust for information transmitted about the virus between TV channels, radio, newspapers, social networks, and news channel sites*.

A Friedman test was conducted to determine whether respondents had a different preference for the five media channels. The results of that analysis indicated that there was a different preference for the five media channels (Chi-Square (4) = 429.667, *p* < 0.001). The effect size as measured by Cohen’s d, was d = 0.12, indicating a very small effect.

A Wilcoxon test was conducted to evaluate whether respondents showed statistically significant differences in trust between media channels grouped two by two. The results indicated a significant difference between trust in TV channels and trust in radio (z = −8.54, *p* < 0.001; the mean of the ranks in favor of TV channels was 163.16, while the mean of the ranks in favor of radio was 150.23; d = SQRT (z^2^/*n*) = 0.28 indicated a small effect size), between trust in TV channels and trust in newspapers (z = −11.40, *p* < 0.001; the mean of the ranks in favor of TV channels was 203.22, while the mean of the ranks in favor of newspapers was 165.6; d = 0.38 indicated a small effect size),between trust in TV channels and trust in social networks (z = −13.95, *p* < 0.001; the mean of the ranks in favor of TV channels was 248.9, while the mean of the ranks in favor of social networks was 162.9; d = 0.46 indicated a small effect size toward a medium effect size), between trust in TV channels and trust in news channel sites (z = −14.93, *p* < 0.001; the mean of the ranks in favor of TV channels was 235.1, while the mean of the ranks in favor of news channel sites was 133.14; d = 0.50 indicated a medium effect size).

The first conclusion after this calculation is that trust in TV channels is dominant compared to any other media channel. The analysis continued to exhaust all pairwise comparisons between the five variables. Thus, we observed a significant difference between trust in radio and trust in newspapers (z = −5.49, *p* < 0.001; the mean of the ranks in favor of radio = 161.1 while the mean of the ranks in favor of newspapers = 141.1; d = 0.18 indicated a small effect size), between trust in radio and trust in social networks (z = −10.42, *p* < 0.001; the mean of the ranks in favor of radio = 237.4 while the mean of the ranks in favor of social networks = 178.2; d = 0.35 indicated a small effect size), between trust in radio and trust in news channel sites (z = −10.77, *p* < 0.001; the mean of the ranks in favor of radio =211.3 while the mean of the ranks in favor of news channel sites = 159.84; d = 0.36 indicated a small effect size).

The second conclusion from this analysis is that trust in radio is dominant compared to trust in newspapers, social networks and news channel sites. Additionally, we observed a significant difference between trust in newspapers and trust in social networks (z = −8.04, *p* < 0.001; the mean of the ranks in favor of newspapers =215.69 while the mean of the ranks in favor of social networks = 167.16; d= 0.27 indicated a small effect size) and between trust in newspapers and trust in news channel (z = −8.11, *p* < 0.001; the mean of the ranks in favor of newspapers = 209.9 while the mean of the ranks in favor of news channel =159.08; d = 0.27 indicated a small effect size).

The third conclusion is that trust in newspapers is dominant compared to trust in social networks and trust in news channel sites. Finally, we observed that there is no significant difference between trust in social networks and trust in news channel sites (z = −0.862, *p* = 0.389 > 0.05).

All the previous data confirm that for the elderly from our sample, trust in TV channels is dominant compared to any other media channels, trust in radio is dominant compared to trust in newspapers, social networks and news channel sites, trust in newspapers is dominant compared to trust in social networks and trust in news channel sites and finally, there is no significant difference between trust in social networks and trust in news channel sites. A possible explanation for these results could be that the channels to which the elderly had most access to were the TV and the radio during the pandemic and thus they trusted more the information they saw on the news or the information they heard on the radio. Furthermore, we observed that the elderly from our sample trusted to a limited extent the information they found on social networks or websites, and this could happen either because they did not use online channels or because they were advised by their relatives or close friends not to trust the information from social networks or websites. Hypothesis 2 (H2) was confirmed in all cases apart from the last case (no significant difference between trust in social networks and trust in news channel sites).

With the data about trust in media channels (the items from question A8), we constructed a statistical index named “trust_media” measuring the confidence in main media channels (TV channels, radio, newspapers, social networks and news channel sites). The new index “trust_media” has good reliability (with Alpha Cronbach = 0.867 > 0.650 for five items). The newly constructed index has the following descriptive values (Table 6).

Starting from this statistical index, we enounced the next hypothesis:

**Hypothesis 3** **(H3).***The level of trust in media channels is significantly different with gender, residency, age category and study*.

A Mann–Whitney U test was conducted and observed that the males and females are not significantly different in terms of trust in media channels (U = 91,127.5, z = −0.690, *p* = 0.490 > 0.05; mean rank: 433.43 < 445.42). The difference between males and females was not statistically significant.

The second analysis revealed the same results: a Mann–Whitney U test was conducted and we observed that the respondents who live in an urban environment are not significantly different in terms of trust in media channels compared to those who live in rural areas (U = 75,311, z = −0.562, *p* = 0.574 > 0.05; mean rank: 443.46 > 432.70). The difference between those who lived in an urban area and those who lived in a rural area is not statistically significant regarding level of trust in media channels.

A Kruskal–Wallis Test was conducted to examine the differences in trust in media channels according to the age categories (1. under 65 years old; 2. 65–70 years old; 3. 70–75 years old; and 4. over 75 years old). Significant differences (Chi-square = 7.942, df = 3, *p* < 0.05) were found among the four categories of ages. A Mann–Whitney U test was conducted and we observed that just the respondents between 65 and 70 years old were significantly different in terms of trust in media channels compared to those with an age over 75 years (U = 30166, z = −2.531, *p* < 0.05; mean rank: 282.44 > 248.01) and the respondents between 70 and 75 years old were significantly different in terms of trust in media channels compared to those with an age over 75 years (U = 19150, z = −1.997, *p* < 0.05; mean rank: 220.77 > 197.25).

Regarding the level of education (school completed), the respondents were not significantly different in terms of trust in media channels (Chi-square = 3.445, df = 3, *p* > 0.05) whatever the level: primary school, gymnasium, high school or faculty. Returning to Table 6, we consider that the level of trust with a mean = 3.43 remains moderate.

Therefore, we observed that Hypothesis H3 is partially confirmed: we do not have statistically significant differences in trust in terms of gender, residency or school completed. Only between some of the age categories, we observed statistical differences.

If the general level of trust in media channels remains moderate, the respondents declared that any information was discussed with other people to verify the sustainability of the information gathered. The people consulted are presented in Table 7.

Of course, the item “asked family members” cannot be a feasible strategy and that is precisely why the call to specialists should have been paramount. Otherwise, 42% would rather justify the conditions of social isolation with the tight regrouping of families in domestic spaces. However, some of the respondents declared that they talked to their family doctor (33%), or that they asked their friends (12%). After some clarification, the personal decisions in the fight against the pandemic followed and the measures taken by the elderly are presented in Table 8.

Thus, most of the respondents decided to get vaccinated (37%), some of them decided to take additional drugs in order to strengthen their immune system (25%), but there were some who decided to not leave the house (18%), to not get vaccinated (10%) or to stop meeting with family members (4%). Overall, we observed that the elderly knew the ways they could protect themselves during the pandemic, most of them resorting to measures such as receiving the vaccine or strengthening their immune system.

#### 5.1.2. The Elderly and the Way of Living during the Pandemic

All these measures taken by the elderly were a recognition of the fact that everyday life had changed consistently for them. In this regard, evaluating the health and mental status was difficult because:-41% of respondents declared that the pandemic situation has had negative effects on their health to a large and very large extent (question B1);-40% of respondents declared that the lack of socialization with people has affected their mental health (question B2);-the emotional states felt most often during the pandemic by the respondents were: fear (39% of the respondents), stress (23%), mental fatigue (16%) and anxiety (14%) (B3);-53% of respondents mentioned that the nationally imposed physical distancing measures changed the way of life they used to have before the pandemic to a large and very large extent (question B4).

In this context, interpersonal communication has become very important, especially for the elderly. Thus, they declared that they communicate 2–3 times a week or daily with children (77% of the respondents), with brothers and sisters (58%), with neighbors (55%) and with friends (52%). The means that they have used most often are presented in Table 9.

Predictably, we noticed that during the pandemic, face-to-face visits decreased significantly (16%) and that communication was mainly made through phone calls (68%). Next, the topics of discussion mostly referred to the virus and these are presented in Table 10.

Hence, the subject that the elderly discussed the most during the pandemic referred to ways of preventing infection with the virus (38%), followed by the severity of the disease (27%) and the difficulties they faced due to the protective measures implemented (13%).

Specific to the elderly in the pandemic condition was the swing between “social life” and “isolation” as an ambivalence that certainly affected mental health, general health and life satisfaction. An online article [153] analyzed the consequences of isolation for the elderly in a pandemic: fear, anxiety and depression. The elderly object to leaving the treatment centers and the isolated ones suffer from the lack of communication/social life: help remains the dedicated telephone lines where the elderly can call. A previous study [56] passes in review some important approaches that can be used to avoid social isolation and loneliness among elderly people: “promoting social connection as public health messaging, mobilizing the resources from family members, community-based networks and resources, developing innovative technology-based interventions to improve social connections, and engaging the health care system to begin the process of developing methods to identify social isolation and loneliness in health care settings”.

In our research, we intended to capture both extremes at the same time: “satisfaction with socialization” and “fear of isolation” because we believe that they can coexist and reinforce each other. In our questionnaire, we introduced the B11 scale question composed of two kinds of items: about socialization (it was easy for me to relate to other people; I had people with whom to share my feelings; it was easy for me to get in touch with other people when I needed it) and about loneliness (I felt isolated/isolated from other people; when I was around other people, I felt isolated/isolated from them; I felt alone and felt like I did not have any friends). With these two kinds of items, we constructed two statistical indices, which are presented in Table 11.

These two indexes have good reliability (“socialization” with Alpha Cronbach = 0.678; “isolation” with Alpha Cronbach= 0.794).

**Hypothesis 4** **(H4).***The evaluation of “satisfaction with socialization” is greater than the “fear of isolation” among the elderly from our sample*.

A Wilcoxon test was conducted to evaluate whether respondents showed statistically significant differences for the “satisfaction with socialization” and “fear of isolation” among the elderly from our sample. The results indicated a significant difference between these two conditions (z = −17.282, *p* < 0.001; the mean of the ranks in favor of “socialization” = 420.77 while the mean of the ranks in favor of “isolation” = 246.69; d = 0.58 indicated a medium effect size). We observed that the “satisfaction with socialization” is superior to the “fear of isolation” and our hypothesis is confirmed.

It was interesting to analyze the correlations that can be calculated between the already defined statistical indicators: “trust_media”, “socialization” and “isolation” (in fact, trust in the media, satisfaction with socialization and fear of isolation). These correlations are presented in Table 12.

We observe a statistically significant correlation between the variable “socialization” and “trust_media” but of low intensity. In other words, trust in the media does not guarantee the possibility of satisfaction with socialization. In turn, isolation and media trust do not correlate.

Calculating the Spearman correlation coefficient (ro) for the variable “socialization” and “isolation”, we deduced that the correlation, although statistically significant, is weak: ro (881) = −0.136, *p* < 0.01. Although we would have expected the two variables to be mutually opposite (so the value of “ro” should be closer to the −1 limit), the intensity of the correlation remained weak, although in the opposite direction. In other words, the “socialization” of the elderly, even in beneficial situations, does not exclude the possibility of isolation and loneliness. It is therefore good to emphasize this sensitive balance between the two extremes.

### 5.2. Results of the Qualitative Research

In our qualitative research, we used a semistructured interview guide and we further present the results of the analysis according to each question that was addressed to the respondents. For the results for each question, we elaborated tables that contain the percentages we mention in the description of the results. All the tables with data that support the information we describe further are presented in Appendix C.

What did the pandemic generated by COVID-19 signify for you?

The pandemic generated by COVID-19 was perceived by the elderly respondents through the prism of restrictions. A percentage of 25% of the respondents identified the pandemic generated by COVID-19 with restrictions; this means, on the one hand, isolation from family and friends, a fact that has a direct impact on the socialization needs of the elderly and implicitly on the quality of life (Appendix C, Table A1). In this context, family relationships were affected, the respondents considering the fact that, due to the particular pandemic context, they benefited from fewer visits from children abroad, which caused them suffering. Thus, the pandemic and implicitly the restrictions applied by the authorities represented, for a quarter of the participants in the study, reduced meetings with family members, which caused them total discomfort. For 15% of the subjects, the pandemic meant restlessness, anxiety and suffering. Moreover, for 15% of the elderly participants in the study, the pandemic meant fear. A total of 10% think that the pandemic has been a difficult time, while another 10% assimilate the pandemic with sadness. Only 5% of respondents consider the pandemic to be insignificant (Appendix C, Table A1). The pandemic period is perceived negatively, being assimilated to excessive information, isolation, limited access to medical services and heightened loneliness.

Who do you think has been the most affected by the COVID-19 pandemic in your community?

The elderly perceive themselves as the most affected category of people by the COVID-19 pandemic; it is the opinion of 65% of respondents. Only 15% consider children to be the most affected category of the COVID-19 pandemic in the community of interest. However, a percentage of 10% of the elderly respondents believe that all vulnerable categories—the elderly, disabled people and children—have been similarly affected by the COVID-19 pandemic. A percentage of 5% consider disabled people to be the most affected, while 5% believe that isolated people are the most affected (Appendix C, Table A2). Therefore, in a significant proportion, the elderly believe that they were the social category most affected by the COVID-19 pandemic in the community of interest for the research. The percentage denotes the vulnerability of the elderly to this pandemic.

Which were the key channels through which you obtained information about the pandemic situation?

Television remains the main channel through which the respondents obtained interesting information about the pandemic. Thus, 95% of subjects considered television as the main channel for obtaining information about the pandemic. Only 5% of respondents join television and the Internet. It is a percentage that shows, as specified, the elderly’s major predilection and appetite for information obtained through television compared to the low percentage of elderly who used the Internet to access information about the COVID-19 pandemic (Appendix C, Table A3). The television station most preferred by respondents is Antena 3, followed by Romania TV. These are the stations from which the information related to the pandemic was taken by the respondents. Other stations watched are Realitatea TV and Digi24.

With whom did you most often talk about the information you saw in the media about the virus?

The most frequent interlocutors of the elderly participants in the study, compared to the information obtained about the virus on mass media channels, were, in a significant proportion, family members. Thus, 60% of the respondents state that they discuss the virus, most often the information obtained through mass media channels, with family members. Next, 25% of the elderly state that they talk to their neighbors, 5% of them talk to their close relatives, 5% also join family friends and 5% join their colleagues’ families (Appendix C, Table A4). Therefore, to a significant extent, the family is the interlocutor of the elderly respondents regarding information obtained from the mass media concerning the virus and the COVID-19 pandemic.

Considering the crisis generated by COVID-19, what were the key problems you have faced, illustrating with a few examples?

The period of the pandemic meant a time course that raised numerous problems. The elderly respondents, participants in the study, were invited to state the main problems they faced. The most significant problems identified by the elderly were, in order, that of restrictions; 50% of respondents believe that restrictions were the most important problem. A significant percentage of 10% consider isolation to be another important problem. Lack of food is considered a problem by 10% of respondents. At the same time, the pandemic period is assimilated with a lack of access to medical services, 10% of respondents considering this a real problem. A percentage of 5% of the respondents are of the opinion that the pandemic period did not raise major problems for the elderly. Another 15% identify the increase in the price of food and utilities, loneliness and caring for grandchildren as problems closely related to the pandemic period (Appendix C, Table A5).

How did you overcome the problems you were confronted with during the pandemic?

The problems faced by the elderly in this study during the pandemic were overcome as follows: 45% of respondents say they have adapted or resigned. The opinion of 20% of the respondents is that they benefited from the support of family, while 10% claim that they benefited from the support of acquaintances. A percentage of 10% claim that the problems they faced during the pandemic have not been overcome and a percentage of 5% claim that it is not appropriate to talk about the problems. Further, 10% of respondents state that the problems they faced during the pandemic were overcome through mental self-education, meditation and phone conversations with family and acquaintances (Appendix C, Table A6). The problems faced by the respondents during the pandemic were overcome from a medical perspective through discussions with the family doctor, through discussions with family and through trust and hope in divinity.

How would you describe your life during this period of the pandemic?

Life during the pandemic generated by COVID-19 is perceived as difficult by 45% of respondents. A total of 20% say they have led a solitary existence, while 10% describe their life as limited and another 10% describe their existence during the pandemic as unsafe. Moreover, 10% of the respondents state that their life during the pandemic generated by COVID-19 was boring and a percentage of 5% thought their life was sad (Appendix C, Table A7). Life during the pandemic generated by COVID-19 is perceived as depressing and generating anxiety, fear, illness and loneliness, a period that affected the elderly through the absence of socialization, through the stress of not contracting the virus, through certain lifestyle habits including eating habits that were radically changed and through loneliness. Overall, the pandemic period represents a tense period for the elderly, with them being too little active, dominated by fear and anxiety.

In the context generated by COVID-19, please mention which were the utilities and basic products whose access was affected in your community.

In the particular context created by COVID-19, 25% of respondents believe that food was the basic product whose access was affected in the community of origin, while 20% believe that medicines had affected access in the community. However, 55% of the respondents, therefore more than half, believe that in the particular context generated by the COVID-19 pandemic, the reference community had no shortages or deficiencies (Appendix C, Table A8). Restricting access to certain utilities also means restricting access to medical treatments, including medical services, surgical interventions, oncological treatments and other types of treatments that benefit people living with chronic diseases throughout their entire lives. At the same time, a main problem faced by the elderly respondents was excessive computerization, with them not being able to keep up with it, and the payment of various bills or utilities could not be made because access to the computer was impossible and the telephones they used were not so efficient.

What kind of medical support was provided to you during the pandemic?

According to the results, 45% of the respondents state that they benefited from medical support, which was provided to them during the pandemic, respectively, they benefited from consultations with the family doctor. A percentage of 5% claim to have benefited from vaccination and a percentage of 5% claim to have benefited from hospitalization. However, 45% of respondents state that they did not receive any kind of medical support during the pandemic (Appendix C, Table A9).

What kind of social support was provided to you during the pandemic?

Regarding the social services provided to respondents during the pandemic, 20% of them state that they have benefited from an emergency or in-kind aid. At the same time, few subjects (5%) claim that they have benefited from free food products and 5% of respondents state that they have benefited from hygiene products at home. A significant percentage of the elderly, represented by 70% of the respondents, claim that they did not benefit from any kind of social support during the pandemic (Appendix C, Table A10).

Who did you spend time with during the quarantine/state of emergency?

Satisfaction evaluation indicators create an image of the mode in which the elderly assess their own life, offering a perspective on the time spent in quarantine, the state of emergency, which meant for 85% of respondents time spent with family, while 15% of the respondents say that they spent time alone (Appendix C, Table A11).

What were the topics you discussed the most with family members during the pandemic?

The topics that were most frequently discussed with family members during the pandemic were mostly topics focused on the pandemic; 65% of respondents claim that the pandemic was the main topic of topics discussed with family members during the pandemic. A percentage of 10% claim that they discussed the consequences of the pandemic and another 10% say that they discussed the price increases that will follow. Moreover, a percentage of 15% discussed loneliness, the effects on the educational system and limited access to medical services (Appendix C, Table A12).

Please describe how the pandemic has affected your family life.

The way the pandemic affected the respondents’ family life is seen as a radical life change by 20% of respondents, 15% of respondents believe that the main consequence of the pandemic was distancing, while 10% consider it to be increased caution and 5% consider loneliness to be accentuated. Half of the study participants claim that the pandemic has not affected their family life in any way (Appendix C, Table A13).

Please describe how your community life has been during the pandemic.

Regarding community relations, respondents were invited to present how life in the community was developed during the pandemic, and 30% state that the pandemic meant both distancing and a lack of social interaction. A percentage of 15% claim that it meant adaptation to new conditions and 15% claim that it meant increased caution. For 15%, the pandemic meant loneliness, for 10% the pandemic signified restrictions and 5% say it was about insecurity. Direct contact with the virus made the elderly more vulnerable, but also the indirect effects of social isolation, embodied in loneliness, limited access to medical services, limited social relationships and reduced accessibility to community life. Only 10% of respondents state that their life in the community during the pandemic had not been affected in any way (Appendix C, Table A14).

What age do you have?

A total of 60% of respondents are between 65 and 70 years old, 25% of respondents are between 71 and 75 years old, 10% of respondents are between 76 and 80 years old and 5% of respondents are between 81 and 85 years old (Appendix C, Table A15).

Gender:

As for gender, 80% of the respondents belong to the female gender, while 20% of the respondents belong to the male gender (Appendix C, Table A16).

Education level (last school graduated).

In terms of education level, 20% of respondents graduated, studied primary school, 5% studied secondary school, 25% studied professionally, 30% studied high school and 20% studied university (Appendix C, Table A17).

The elderly person lives alone in the household.

A total of 5% of respondents are people who live alone in the household, while 25% of respondents do not live alone in the household (Appendix C, Table A18).

The elderly person lives with the family.

A total of 5% of respondents live with their family, 75% of the respondents do not live with their family (Appendix C, Table A19).

The environment of origin:

A total of 95% of respondents come from an urban environment and 5% of them come from rural areas (Appendix C, Table A20).

## 6. Discussion and Conclusions

In the context of the pandemic, one of the categories that was most affected by the virus and by the measures implemented in order to stop the spread of the virus was represented by the elderly. In this regard, similarly to previous studies [26,27,28], we argue that the older generation was affected by the virus to the highest degree compared to other vulnerable groups due to their risk of being isolated in order to protect their health.

The purpose of our paper was to analyze the ways in which mass media and very often contradictory information flows influenced the life and personal communications of the Romanian elderly during the COVID-19 pandemic. In this regard, in our paper, we were interested in identifying the media channels used by the Romanian elderly to get information about the pandemic, identifying the main types of information followed by the elderly on the media communications channels, assessing the trust of the elderly in certain media channels, identifying the measures the elderly took in order to protect themselves during the pandemic, identifying the perception of the elderly about the way the pandemic affected their health and emotional state and identifying the ways in which the social relations of the elderly were affected during the pandemic.

Considering the results of the quantitative research, most respondents described the pandemic period using negative words such as fear, disease, quarantine, disaster and virus. In the context of the channels they most used during the pandemic, the elderly declared that they watched television and some of them also accessed the Internet or listened to the radio. Thus, our results show that there were differences in the preferences of the elderly for different types of media channels, but we highlight that the effect size of the statistical analysis conducted is rather small. In this regard, while looking for the factors that influence the use of media channels among the elderly, the results revealed that gender and residential environment (urban/rural) do not influence the way the elderly use mass media channels. However, the research shows different results according to the last school graduated and to the age of the elderly. Thus, the elderly who had graduated from high school and who were between 65 and 70 years old were the ones who had the highest level of media consumption compared to the other categories. Furthermore, it should be noted that at the beginning of the pandemic, the Romanian Government issued an Emergency Ordinance (no. 63 of 7 May 2020) “for organizing and conducting public information campaigns in the context of the epidemiological situation caused by the spread of COVID-19”. Further analysis of the distribution of the important funds (over EUR 40 million) was made by the Expert Forum [154]. In this report, it is specified that some political inferences were possible because “the period in which all the funds were promised overlapped with the electoral period”. Other observations refer to the “allocation criteria, the presence on the list of some beneficiaries with significant amounts of highly politicized providers” and the observation that “the beneficiary, who received significant funds from the Government, becomes vulnerable to the perspective of journalistic independence”. In the same report, it is specified that “the first 10 suppliers collected 50% of the money promised by the Government. Most of the money went to national television, but also to several online portals”. All TV channels mentioned received a great proportion of money even if some of these channels “promoted at the same time the messages sent by the government and conspiratorial themes or in conflict with the official data”.

Taking into account the type of information that the elderly mostly saw on the media channels they followed and the frequency of that information, the respondents declared they mostly saw information about the rules they had to follow, about prevention methods, about the number of the victims and about ways to cure the disease. What was interesting was the number of respondents who declared they saw the news about the negative effect of the vaccine on the channels they watched. These results could thus be an indicator of the amount of fake news that was spread on media channels regarding the virus and the vaccine.

Given the trust in media channels, the results showed that the respondents declared that they had the most trust in TV channels compared to any other media channel. Then, trust in radio registered higher numbers compared to trust in newspapers, websites and social networks. Furthermore, trust in newspapers registered higher numbers compared to social networks or websites. Thus, we observed that the respondents had little trust in Internet sources and this could happen either because the elderly do not frequently access Internet sources or because they were advised by their friends or relatives not to trust the information they see on social networks or websites. In this context, we highlight that when we compared the trust in media channels, the effect sizes were: small in the context of comparing: TV channels and radio, TV channels and newspapers; small to medium in the context of comparing TV channels and social networks; and medium in the context of comparing TV channels and news websites. Furthermore, the effect sizes were small in the case in which we compared trust in radio and newspapers, when we compared trust in radio and social networks and when we compared trust in radio and news websites. Even though trust in newspapers was higher than trust in social networks and news websites, the effect size of the statistical analyses was small.

Moreover, in terms of factors influencing trust in media channels, the results showed no difference in the opinions of the elderly depending on their living environment (rural, urban) or their educational level. However, the research showed that the trust in media channels of people aged between 65 and 70 years old was significantly different compared to those aged between 70 and 75 years old, meaning that the first category tended to trust media channels to a higher extent than the second category.

Given the social relations of the elderly in the context of the pandemic, the results showed that the people with whom they talked the most in this period were their children, followed by their spouses. Moreover, when discussing the pandemic and the virus, most respondents declared they chose to discuss it with their family members, but a large amount of them mentioned they would rather talk with their family doctors. In this regard, we observed that the elderly mostly preferred to receive advice from their family members and their doctors.

In terms of the coping mechanism to which the elderly resorted in order to protect themselves from the virus and survive during the pandemic, most respondents decided to get vaccinated or to take additional drugs in order to boost their immune system. However, some of them decided not to leave their house or not to meet their family members. Hence, we observed that most respondents had positive attitudes towards the vaccine and they resorted to appropriate measures in order to protect themselves. A possible explanation for this result would be that, since the TV was the channel that the elderly trusted the most, they may have been influenced by the information spread by TV channels regarding the benefits of the vaccine or the benefits of certain additional drugs.

When communicating with their family members, the elderly mostly resorted to phone calls and they used very little communication through video calls or online platforms. In this regard, it is possible that the elderly had little access to Internet sources or mobile phones that could have allowed them to communicate with their families through certain apps. In this regard, our paper is in line with a previous study [24], which highlighted the role of technology in maintaining social interaction and communication in crisis periods.

The topics that the elderly mostly discussed with their families referred to ways of preventing infection with the virus, the severity of the disease, the difficulties they encountered and their physical and mental health.

Considering the qualitative research, the results are in line with the results of the quantitative research in the sense that the elderly described the pandemic with negative words, highlighting that during this period, they felt isolated and lonely because they could no longer meet their family, and respondents also stated that the main channel they used in order to obtain information about the virus was the TV.

Moreover, the elderly perceived themselves as the group category that was most affected by the pandemic. In other words, the respondents understood that the virus is more likely to severely damage their health compared to other groups of people. As the results of the quantitative research stated, the results of the qualitative study showed that the elderly mostly talked with their family members about the virus and ways of preventing infection.

When it comes to the problems faced, for most respondents, the restrictions they had to comply with were the main problem. Some of them declared that the biggest problem was loneliness, the inability to be with their families or the lack of access to medical services.

Taking into account the coping mechanisms of the elderly, most of them declared they tried to adapt to the new way of life and that they received help from their friends and families. Others resorted to methods such as mental health education or meditation. In the context of their health issues, the elderly managed to get through the pandemic with the help of their doctors and through hope and trust in divinity. Thus, similar to previous research [75], our study also shows that the elderly can resort to diverse coping mechanisms in times of crisis.

In the context of the lifestyle during the pandemic, the elderly described the period as full of fear, anxiety and difficulties. The elderly were afraid of the virus, they could not communicate with their family members and they had to change their habits due to the restrictions. Furthermore, they declared that they had issues with access to food, medication and public services. In other words, the elderly had trouble with buying food and medicine and paying their bills due to excessive digitization and the lack of proper devices or digital skills. Taking into account the difficulties the elderly faced during the pandemic, our study is in line with previous studies [48,49], which showed how the pandemic and the rules implemented affected social activities and the mental health of people. Moreover, similar to a previous study, our research also emphasizes the fact that the pandemic made people worry more about their life [50] and that it generated feelings of fear [51] or anxiety [52].

However, when asked how the pandemic influenced their family life, most respondents declared that the pandemic did not affect their family life. This might mean that, even if they were no longer able to meet face to face their relatives, the ties the elderly had with them remained the same and they were not negatively influenced by the pandemic.

Considering the medical and social services, the elderly declared that they received medical support from their family doctors but a large amount of them stated that they did not receive any kind of medical or social support.

Therefore, through our study, we managed to obtain a perspective regarding the way information from mass media affected the life of the elderly during the pandemic and regarding the difficulties they faced in this time of crisis. Hence, the mass media did not exercise its surveillance function during the pandemic, the elderly declaring that they saw on TV and other channels both information about prevention methods and negative information about the vaccine. Thus, the information spread by media channels was sometimes contradictory and generated feelings of fear and anxiety among the elderly. Furthermore, based on the results of our research, we argue that the children of the elderly also had a role in influencing the behavior of the elderly and the ways this vulnerable group decided to take measures in order to cope with the pandemic situation.

Hence, we argue that mass media channels have an essential role during periods of crisis and media specialists and professionals from other fields should take into account this matter. In this regard, a previous study highlighted the importance of the media channels by showing that the COVID-19 pandemic has influenced the adoption of telemedicine in the healthcare sector and that the evolution of the pandemic affected the subject discussed on Twitter by the Spanish community and implicitly affected the emotions associated with those subjects [155]. Furthermore, the importance of studying mass media use in the context of older adults was emphasized by a previous study that focused on the adoption of social networking sites by older adults and that identified several types of elders who use these kinds of sites [156], as well as by a previous study that showed how older adults accept and use social networking sites after the pandemic period [157].

From the viewpoint of the theoretical implications, this paper contributes to the literature regarding the coping mechanisms to which the elderly resorted to during the pandemic and regarding the impact that the pandemic had on the lifestyle of the elderly. From the viewpoint of the practical implications, the results obtained through our research show the media consumption of the elderly and the struggles they faced during the pandemic period and such results could be considered a point of reference for the way the lifestyle of the elderly could be improved in general and in times of crises in particular.

### Limitations and Future Research Directions

In terms of the limitations of the paper, one limitation is represented by the fact that the study has only been conducted in Romania. In this regard, in a future study, the research instruments should be used in order to also gather information from people who live in other countries. Another limitation is represented by the fact that the data were collected during the pandemic and thus future similar research should be conducted in order to see if and how the responses of the elderly differ. Furthermore, another limitation is represented by the fact that the study was only conducted on one vulnerable group (the elderly) and future research should take into account other vulnerable groups such as people with disabilities in order to make a comparative analysis between these groups. Moreover, future research should take into consideration the development of a comparative analysis of the mass media experiences of the group of older adults and the experiences of other categories of people affected by crises. Another element that should be taken into consideration in a future similar study would be the use of theories, such as the elaboration likelihood model (ELM).

## Figures and Tables

**Figure 1 healthcare-11-01816-f001:**
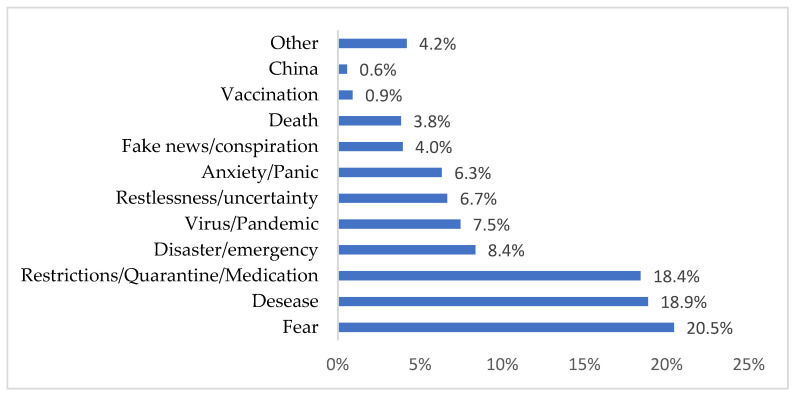
A single word that characterizes the pandemic period.

**Figure 2 healthcare-11-01816-f002:**
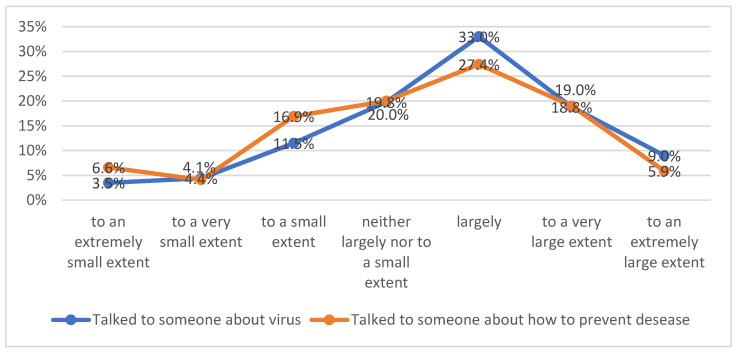
Talking about the virus and how to prevent disease.

**Table 1 healthcare-11-01816-t001:** The structure of the sample.

Variables	Categories	Percentage
Residency environment	Urban	72.5%
	Rural	27.5%
Living place during the pandemic	House	50%
	Apartments	50%
Gender	Male	41%
	Female	59%
Age category	Under 65 years old	16.6%
	65–70 years old	36.2%
	70–75 years old	22.6%
	Over 75 years old	24.6%
School completed	Primary school	12%
	Gymnasium	21%
	High school	49%
	Faculty	18%

**Table 2 healthcare-11-01816-t002:** Percentages of the 65–74-year-old population in Romania regarding Internet activities.

Activity on the Internet	Percentage
Telephoning/video calls	35.7%
Social networks	34.2%
Finding information about goods and services	20.9%
Internet banking	4.3%
Seeking health information	19.5%

Source: Eurostat, 2022. (online data code: isoc_ci_ac_i).

**Table 3 healthcare-11-01816-t003:** How often have you seen the following types of information on the communication channels you have followed?

	The Percentage for “Often and Very Often”
Information about the rules/measures you must follow	82%
Information on how to prevent the virus	80%
Information on the number of victims	79%
Information about the number of infected people	74%
Information on the positive effects of the vaccine	68%
Information on ways to cure the disease caused by the virus	59%
Information about the negative effects of the vaccine	48%

**Table 4 healthcare-11-01816-t004:** Vaccination decision and exposure to positive/negative information about COVID-19.

I Read Positive Things about Vaccine			
Rarely	Neither Rarely nor Often	Often	Total
54	44	228	326
I read negative things about vaccine			
115	50	161	326

**Table 5 healthcare-11-01816-t005:** People you have talked to most often about the virus.

	Percentage
Children	42%
Spouse	23%
Other family members	13%
Friends	10%
Siblings	5%
Neighbors	4%
Someone else	3%

**Table 6 healthcare-11-01816-t006:** Descriptive statistics for the “trust_media” index.

	*n*	Minimum	Maximum	Mean	St. Dev.
Trust_media		1.00	7.00	3.43	1.29
Valid *n* (listwise)	881				

**Table 7 healthcare-11-01816-t007:** Ways to verify information about the virus.

	Percentage
Asked family members	42%
Consulted with the family doctor	33%
Asked friends	12%
Asked neighbors	4%
Other people	9%

**Table 8 healthcare-11-01816-t008:** Decisions made to fight the pandemic.

	Percentage
Get vaccinated	37%
Take additional drugs for immunity	25%
Stop leaving the house	18%
Not get vaccinated	10%
Stop meeting with family members	4%
Something else	6%

**Table 9 healthcare-11-01816-t009:** Means of communication.

	Percentage
Phone calls	68%
Face-to-face visits	16%
Video calls through mobile applications	13%
Online platforms	2%
Other	1%

**Table 10 healthcare-11-01816-t010:** Topics of discussion during the pandemic.

	Percentage
Ways to prevent the virus	38%
The severity of the disease	27%
The difficulties you face due to implemented protection measures	13%
Your physical health	12%
Your mental health	10%

**Table 11 healthcare-11-01816-t011:** Descriptive statistics for two indexes.

	*n*	Minimum	Maximum	Mean	Std. Deviation
Socialization	881	3.00	15.00	10.43	3.222
Isolation	881	3.00	15.00	6.96	3.481
Valid *n*	881				

**Table 12 healthcare-11-01816-t012:** Spearman correlations of the defined indexes.

Variables	*n*	M	SD	1	2	3
1. Trust_media	881	3.43	1.29	-		
2. Socialization	881	10.43	3.22	0.085 *	-	
3. Isolation	881	6.96	3.48	0.045	−0.136 **	-

* Correlation is significant at the 0.05 level (2-tailed). ** Correlation is significant at the 0.01 level (2-tailed).

## Data Availability

The data presented in this study are available in this article.

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
