# Peer review of "The Role of Mass Media in Influencing the Lifestyle of the Elderly during the COVID-19 Pandemic"

_healthcare, 2023, doi:10.3390/healthcare11131816_

Round 1

Reviewer 1 Report

Dear authors, I'm glad to have the opportunity of reviewing your manuscript, which has relevant topic for public health and psychology. The purpose of the research was to analyze the ways in which mass media and very often contradictory information flows influenced the life and personal communications of Romanian elderly during the COVID-19 pandemic. Manuscript has its’ strengths, however, there are my comments that I believe would help to increase the quality of the manuscript.

-        Abstract: The abstract does not follow the editorial standard indicated for IJERPH: “The abstract should follow the style of structured abstracts, but without headings”; therefore, remove the words: Background, Methods, Results and Conclusions.

The abstract should provide information about study composition (gender, age).

-        The introduction is too long and confusing to highlight the research gap. Subsection with information about examined variables should be included. You go from coping to resilience and then back to coping again. It is extremely confusing. This paper needs to be carefully reorganized with a clearly stated argument spelled out in the opening pages.

-        In the introduction you wrote: "On November 2, 2021, Romania was the second least vaccinated state of the European Union, despite the fact that a record number of daily deaths from the coronavirus were reported and there were no intensive care beds available in Romania." You should give some information about the actual situation.

-        The Purpose and objectives of the research and Hypotheses of the research are described under Materials and Methods. These must be described under its own heading. The objectives are extensive and very wordy. Please be consistent.

-        The Method section lacks information regarding study design, recruitment process and statistical power (sample size calculation). Eligibility of the population is missing, have you performed a regression study? How did the authors recruit the participants, where? Individually or in groups? Please expand and provide some more details in the area of intervention.

How many were invited and how many declined participations?

How many were excluded and due to what exclusion criterion?

-        I must necessarily recommend a profound revision of the methodology which is reductive and opaque, especially for statistical analysis.

-        The choice of measures is not very clear. Why you didn't use validated scales?

-        Move this part in The research instruments

“The analysed variables were: a word describing the mental 439

representation of the term Covid 19, most followed media channels, most followed  tele- 440

vision channels, types of information followed, the frequency of discussions about spread 441

the virus, the frequency of discussions about disease prevention,categories of interlocu- 442

tors about the pandemic, the level of confidence in diverse media channels, forms of vali- 443

dation of the information, decision made after the media exposure, effect of the pandemic 444

on health state, effect of the pandemic on mental state, emotional states during pandemic, 445

effect of distancing measures, categories of relatives in direct contact, intensity of commu- 446

nication with relatives, types of channels of interpersonal communication, pandemic re- 447

lated subjects of communication, intensity of social relationship, intensity of necessary 448

support received, the quality of family ties, the tendency to avoid social contacts, the in- 449

tensity of the feeling of alienation, main activities during the pandemic, main life aspect 450

affected, type of living, possession of a dog, the level of satisfaction with the vaccine, 451

reasons for vaccination, reasons of non-vaccination, main challenges during the pandemic.”

-        Figure 1. A single word which characterizes the pandemic period   THIS IS FIGURE 2

In my opinion Figure 2 is useless, you can delete it.

-        Table 2, Table 4 and Table 7: delete these tables, you may briefly describe the findings in the text.

-        A zero should not be inserted before a decimal fraction when the number cannot be greater than 1. For example, p < 0.05 should be written as “p < .05.” Continues in the same way!

Typically, if the exact p value is less than .001, you can merely state p < .001.

-        Please print the effect size category cut offs used.

-        General comment for results/discussion: effect sizes never get discussed.

-        Use past tense when discussing the procedure and results as well as other researchers' procedures and results.

-        English needs extensive editing to facilitate ease of reading.

-        Where is Institutional Review Board Statement??? Please add number.

    English needs extensive editing to facilitate ease of reading.

Reviewer 2 Report

The introduction needs to clearly lay out the thesis and purpose of the paper. It is very difficult to ascertain the overall intent of the work.  The background lacks clarity and focus. This lack of concentration distracts from the quality of the work and makes the piece very hard to follow or evaluate. The data collected represents several thematic analyses that could be explored over multiple papers. As presented, this work is very difficult to comprehend.  There are also unclear transitions between paragraphs making transitions from one paragraph to the next quite jarring. More detailed comments can be found below about the contents of the article. 

I am not sure the upward curve has been demonstrated to be related to quality of life

You have do define how you are using the term quality of life

Line 70 – 74 is contradictory. You state that illness and death was highest in persons 50-59 but 85% of mortality was in persons 60+

Line 303 to 304 – you need to directly tie restrictions to the quality-of-life factors you have introduced. Your description of the restrictions (85-99) are not very comprehensive.

While your paper includes a lot of rich information, it is important to only include the details that help support your readers understanding of your work. I found a lot of the detail to be superfluous. For example, your inclusion of coping did not seem overly relevant to your central thesis.

I strongly recommend a complete overhaul and edit of the introduction section of your paper.

Materials and Methods

You need to clearly outline you inclusion, exclusion criteria. Inclusion of 45 year-olds is not common practice among research relating to older persons. The lower limit is usually 55.

Q4 – restate, I am not sure what you are establishing here.

How do Q6 and Q7 relate back to your research aim?

I feel as though you have two papers worth of material based on your survey data. The first submission could be focused on the relationship between published and social media. The other is rooted in informal networks.

489 – in your opinion, should be related to research findings  

540 – 555 – this should be in the discussion section not results.

Your results demonstrate some imbedded bias on behalf of the researchers. For example, you correlate negative effects of the vaccine with the spread of misinformation and fake news (562-563) however vaccines always carry with them negative effects (from minor to severe). Additionally, you illude to a high prevalence of contradictory information however this has yet to be established in your data, literature review, or examples.

629 – 641. You need to root your explanation of your findings in the data (e.g., pull from your open-ended responses or interviews). Your text reads as though it is the opinions of the authors. Additionally, you are treating all of your media as uniform by type, is this appropriate (e.g., are all television stations equally established?)

745-746 – are satisfaction with socialization and fear of isolation dichotomous terms. You also have to be able to demonstrate a causal relationship – linking isolation to COVID and not other underlying issues.

What was the link between the qualitative and quantitative studies?  Where the survey results analysed to help structure the interview guide

For qualitative results you should report your N along with the percentage

799 – how did you, or your participants, define effected?

Your qualitative data should include direct quotes from your interviews to support your analysis

For your qualitative data, you do not need to report on all of your findings, only those relevant to your thesis.

Your reporting on demographic data 934 – 950 often do not tally to 100%, where are the missing respondents? You have also not presented any information on the actual number of respondents.

Your introduction and background include a lot of information about quality of life however this does not appear to have been a focus of your research. Perhaps this emphasis should be removed from the introduction.

Reviewing the interview guide, it does not really link directly to the use of media for the purposes of information. I wounder if its inclusion adds robusticity to this work.

There are many structural challenges throughout this work likely due to language conventions (e.g., direct translation from one language to anther often results in grammatic and syntax errors that negatively impact understanding). Also naming conventions (e.g., terms like study (line 383), in English often translated to level of education) also distract from comprehension.  

Author Response

Please see the the attachment.

Reviewer 3 Report

This study contributes to our understanding of the impact of mass media and contradictory information on the lives of the elderly population in Romania during the COVID-19 pandemic. The findings reveal a range of negative emotions experienced by older people, including fear, loneliness, and anxiety. Television emerges as their primary and trusted source of information. While they are aware of protective measures, they face challenges related to loneliness and limited access to medical services. Furthermore, the consumption of mass media appears to influence their behavior and may contribute to feelings of anxiety.

It is essential to acknowledge that the findings of this study may have limited generalizability to other regions or populations due to cultural and contextual differences. Additionally, the study is determined by a need for a long-term perspective and reliance on self-reporting, which introduces the potential for bias.

Nevertheless, the study offers several noteworthy contributions. Firstly, it provides valuable insights into older people's experiences and challenges during the pandemic. The study sheds light on their unique circumstances by exploring their emotions, communication patterns, coping mechanisms, and perceptions of media channels.

Secondly, adopting a mixed-method approach, combining quantitative surveys and qualitative interviews enhances the study's comprehensiveness. This approach allows for a more holistic understanding of the topic, capturing numerical data and individual narratives.

Furthermore, identifying television as the most trusted source of information for older people has practical implications. This finding can inform communication strategies during crises, ensuring accurate and helpful information reaches older people through their preferred media channels.

The study also highlights the resilience and adaptability of the elderly population. Despite their challenges, they employ various coping mechanisms such as vaccination, seeking support from family members and doctors, and adapting to new circumstances. These positive coping strategies demonstrate their ability to navigate difficult situations and safeguard their well-being.

Lastly, the research has practical implications for improving the quality of life for older people. The insights gained from this study can inform strategies to address their specific needs and concerns, promote social support, and ensure access to essential services.

In short, while the study has limitations, it provides valuable insights into the experiences of older people during the COVID-19 pandemic. In addition, it contributes to our understanding of the impact of mass media and offers practical implications for supporting the well-being of older people during crises and in general.

Some additional comments

Research Question:

The primary research question addressed in this study was to analyze the influence of mass media and often contradictory information flows on the lives and personal communications of elderly Romanian individuals during the COVID-19 pandemic. The aim was to understand how media consumption patterns affected the elderly population in Romania during the crisis.

Originality and Relevance:

The topic of investigating the impact of media on individuals is extensively studied; however, the focus on older adults, particularly during times of crisis such as the COVID-19 pandemic, remains relatively understudied. Therefore, this research addresses a specific gap in the field by examining the unique experiences and responses of elderly individuals to media during a global crisis.

Contribution to the Subject Area:

This study contributes new data and insights to the existing body of knowledge on the effects of media consumption, specifically in the context of older adults. By examining the experiences of Romanian elderly individuals during the pandemic, it provides valuable information that can enhance our understanding of the media's role in shaping the perceptions and behaviors of this demographic.

Methodological Improvements and Further Controls:

The study's descriptive methodology should be acknowledged when evaluating the research findings. Collecting data from older adults can be challenging, and the researchers have made a commendable effort to obtain the necessary information.

For future studies, comparing the sample of older adults with other generational cohorts affected by crises and examining their media experiences is recommended. Additionally, employing theories such as the Elaboration Likelihood Model (ELM) can enable researchers to abstract from specific media channels and explore the effects of media on different age groups.

Consistency of Conclusions:

The conclusions drawn in the discussion section align with the evidence and arguments presented throughout the study. For instance, the study successfully addresses its proposed objectives, such as identifying the media channels used by elderly Romanian individuals during the pandemic and evaluating their trust in these channels. The findings indicate that TV channels were the most trusted source of information for the elderly participants, surpassing other media channels.

Appropriateness of References:

The references provided are suitable for supporting the research topic. However, if a broader understanding of the impact of the COVID-19 pandemic through media is desired, the following studies may be worth considering:

         Gaitán, J. A., & Ramírez-Correa, P. E. (2023). COVID-19 and telemedicine: A netnography approach. Technological Forecasting and Social Change, 190, 122420.

Furthermore, if a comparative analysis of media usage by older adults during a pandemic in developing countries is of interest, the following studies could be reviewed:

         Ramírez-Correa, P. E., Arenas-Gaitán, J., Rondán-Cataluña, F. J., Grandon, E. E., & Ramírez-Santana, M. (2023). Adoption of social networking sites among older adults: The role of technology readiness and generation in identifying segments. Plos One, 18(4), e0284585.

         Ramírez-Correa, P., Grandón, E. E., Ramírez-Santana, M., Arenas-Gaitán, J., & Rondán-Cataluña, F. J. (2023). Explaining the Consumption Technology Acceptance in the Elderly Post-Pandemic: Effort Expectancy Does Not Matter. Behavioral Sciences, 13(2), 87.

Additional Comments on Tables and Figures:

While the tables provide necessary information, some improvements can be made to enhance their clarity. For example, adding column headers to all tables (Table 2, 5, 6, 8, 9, 10, 11, 12, 13, and 14) and formatting them according to the journal's standards will improve their presentation.

Round 2

Reviewer 1 Report

The revised manuscript is suitable for the submission 

Author Response

We are very grateful to the reviewer for the time spent analyzing our revised manuscript and for all the useful suggestions that the reviewer provided in the first round of review. Thank you!